# Benign or Low-Grade Malignant Masses Occupying the Pelvic Canal Space in 11 Dogs

**DOI:** 10.3390/ani11051361

**Published:** 2021-05-11

**Authors:** Erica Ilaria Ferraris, Davide Giacobino, Selina Iussich, Matteo Olimpo, Alberto Valazza, Marina Martano, Paolo Buracco, Emanuela Maria Morello

**Affiliations:** 1Department of Veterinary Sciences, University of Torino, Largo Braccini 2, 10095 Torino, Italy; ericailaria.ferraris@unito.it (E.I.F.); davide.giacobino@unito.it (D.G.); selina.iussich@unito.it (S.I.); matteo.olimpo@unito.it (M.O.); alberto.valazza@unito.it (A.V.); paolo.buracco@unito.it (P.B.); 2Department of Medical-Veterinary Science, University of Parma, Strada del Taglio 10, 43126 Parma, Italy; marina.martano@unipr.it

**Keywords:** dog, intra-pelvic mass, tenesmus, colorectal and urethral compression

## Abstract

**Simple Summary:**

Large canine pelvic masses, even though benign, can seriously affect dogs’ lives, causing problems regarding defecation and urination, and sometimes complete rectal and/or urethral obstruction. Many of these patients are euthanized owing to their poor clinical condition as a result of chronic compression or because their disease is erroneously considered untreatable. The aim of this study was to evaluate the clinical data of dogs with intra-pelvic benign or well-differentiated malignant masses referred to the Torino Veterinary Teaching Hospital, and treated surgically. Clinical signs, diagnostic approach, surgical procedures and outcome were evaluated. The majority of the dogs recovered uneventfully from surgery in a couple of days, with rapid resolution of the severe preoperative clinical signs. All the dogs experienced long survival with no disease recurrence or progression, even in the well-differentiated malignant tumors. Clinical findings, especially those obtained using digital rectal and vaginal exploration are mandatory for assessing the presence of the tumor and its relationship to the other pelvic structures. Currently, in the authors’ opinion, preoperative computed tomography (CT) is also highly recommended, even if this procedure was not performed in 4 out of the 11 dogs.

**Abstract:**

Dogs with benign intra-pelvic rectal or vaginal masses show symptoms indicating compression on the adjacent organs. Clinical signs usually develop late when the lesion is large enough to interfere functionally. The dogs were referred for severe fecal and/or urinary tenesmus. The data collected included signalment, clinical signs, results of physical examination, pre-surgical diagnostic tests, surgical technique used, surgical complications and histological findings. Digital rectal and vaginal examination allowed the detection of a mass occupying space in the pelvic cavity in all patients. Abdominal ultrasonography and/or total body computed tomography (CT) were used to better characterize the lesion and to exclude a metastatic spread of the tumor in case of malignancy. A dorsal approach to the rectum, a dorsal episiotomy, a midline celiotomy, and a combined perineal and abdominal approach were performed to remove the mass. No postoperative complications were observed. Benign and well-differentiated malignant mesenchymal neoplasms were histologically diagnosed. As a consequence of the chronic urethral compression caused by the mass, urinary incontinence and/or urinary retention were observed for a few postoperative days. Fecal tenesmus resolved in all cases in the immediate postoperative period. The dogs’ quality of life quickly improved after surgery, especially considering the serious and life-threatening pre-surgical clinical conditions. Both the recovery time after surgery and overall survival were also evaluated.

## 1. Introduction

Pelvic structures, such as the distal part of the descending colon, rectum, vagina, sacral lymph nodes and prostate can be affected by tumors which, as a result of their increased size, may cause a reduction in the pelvic space, thus determining fecal and urinary problems. The behavior of these tumors can be either benign or malignant. Treatment choice and prognosis vary depending on tumor histotype as well as on the oncologic outcome. Cytology or histology from bioptic samples, imaging techniques for staging purposes and/or preoperative planning are usually recommended.

Benign or low-grade malignant intra-pelvic tumors are rare in clinical practice [1] and, for the most part, have a rectal or a vaginal origin. Although almost 60% of rectal tumors are malignant, with a high local metastatic rate and local invasiveness, with adenocarcinoma being the prevalent histotype [2], the majority of the space-occupying lesions of rectal origin are benign with leiomyomas, fibromas and fibro-leiomyomas being the most frequent histotypes [1]. Leiomyoma occurs more commonly in the stomach; however, it has also been reported in the small intestine and colon-rectum [3]. Vaginal tumors are rare in dogs and the majority of them are leiomyomas [4]. Middle-age to older intact females are most commonly affected. Intra-pelvic benign or low-grade malignant tumors can also arise from neural, vascular, glandular and adipose tissue [5].

Patients with benign or low-grade malignant intra-pelvic masses may show signs resulting from the compression of the adjacent organs, such as the rectum and the urethra. Due to the slow growing, non-invasive and non-metastatic behavior of these masses, clinical signs usually develop late in the course of the disease, i.e., when the lesions are large enough to interfere functionally. The dogs are thus presented for tenesmus (fecal and urinary) and constipation.

In some instances, the lesion may cause complete occlusion of the urethra and, less frequently, of the colon-rectum, thus creating a clinical situation not compatible with life [4].

Older dogs are more commonly affected, and poor clinical conditions may sometimes be evident. This, along with the presence of a large pelvic mass, believed to be unresectable, may lead to poor prognostic judgment and to euthanasia.

Due to the non-invasive behavior of these tumors, a marginal excision is often feasible and curative. Episiotomy or a dorsal approach to the rectum are the most reported surgical techniques [6,7]. The removal of the mass provides complete relief of the clinical signs, with rapid improvement of the patient’s condition. If the mass extends into the abdomen, a caudal abdominal approach may also be necessary.

Of the diagnostic examinations which help to differentiate between benign and malignant intra-pelvic tumors, digital rectal examination (DRE) should be utilized. It is a simple examination which allows discovering the presence of a pelvic mass and defining its characteristics, such as consistency, mobility and localization.

In the case of a well-defined, compact mass, dorsal or ventral to the rectum at DRE, a benign rectal or vaginal mass can be suspected.

The aim of this retrospective study was to support the hypothesis that a simple DRE (combined or not with vaginal digital exploration) may provide important information regarding the non-invasive nature of the lesion. Nevertheless, a preoperative diagnosis should always be attempted. The cytology in case of consistent mesenchymal lesions dorsal or ventral to the rectum is often not diagnostic but helps in differentiating these lesions from an epithelial or round cell lesion, more prone to exfoliate [8]. A biopsy for histological examination could be done in the case of non-diagnostic cytology. Imaging techniques, such as computed tomography (CT) may be also used in diagnostic planning.

The data of the dogs referred to the Torino Veterinary Teaching Hospital (VTH) for intra-pelvic histologically benign or low-grade malignant masses of rectal or vaginal origin responsible for severe functional problems in defecation and/or urination were reviewed. Diagnostic procedures, type of treatment, postoperative complications and recovery time were evaluated.

## 2. Materials and Methods

The medical records of dogs referred to the Torino VTH from January 2002 to December 2020 for fecal and urinary partial or complete obstruction were identified and reviewed. The inclusion criteria were the presence of a well-defined compact mass, dorsal or ventral to the rectum, clinically evaluated at DRE and surgically excised. Only benign or low-grade/well-differentiated malignant tumors were included.

Tumors localized in different positions than those dorsal or ventral to the rectum at DRE and arising from anatomical structures other than rectum and vagina were excluded.

Written consent was obtained from the owners for the anesthetic, diagnostic, histological and surgical procedures before proceeding. They signed an informed consent for the procedure as a standard procedure for the surgical treatment of AGASACA tumor. No specific informed consent statement was signed for the inclusion in this retrospective study. Perioperative standard-of-care management, including analgesia, was assured for all the dogs. The study did not fall within the application areas of Italian Legislative Decree 26/2014 which governs the protection of animals used for scientific or educational purposes; therefore, ethical approval was waived for this study. The animals were not treated as part of an experimental study; however, out of necessity, only the data regarding the study were later selected and included in this study.

The data collected included signalment, clinical signs, results of physical examination including digital rectal and vaginal exploration, pre-surgical diagnostic tests, surgical technique used, surgical complications and postoperative histological findings. The disease-free interval (DFI), defined as the time between surgery and local recurrence or metastatic spread for malignant tumors, and the survival time (ST), considered to be the time between surgery and the dog’s death for any reason, were calculated. Dogs with a follow-up time shorter than 365 days were excluded from the DFI and the ST evaluations. Recovery time, considered to be the return to normal defecation and/or urination, was also evaluated.

The work-up generally included blood cell count, serum chemistry profile, 3-view thoracic radiography and abdominal ultrasonography, or a total body CT scan.

Digital rectal examination (plus vaginal examination in the case of a mass positioned ventrally to the rectum) was carried out, thus allowing the detection of a mass occupying space in the pelvic cavity and a useful characterization of the mass, i.e., tumor localization, adhesion to the underlying structures and consistency.

A cytologic (fine needle biopsy) or a histologic (tru-cut needle) sampling was attempted using a transcutaneous perineal, transrectal or an transabdominal approach, ultrasound guided, as opportune.

Due to the long study period, different analgesic and anesthetic protocols were used; antibiotic prophylaxis was administered at anesthesia induction and then every 90 min until completion of surgery.

For a dorsal rectal approach, the dogs were placed in sternal recumbency with the tail bandaged, reflected and secured cranially; soft padding was placed between the table edge and the cranial thigh in order to avoid any injury.

Anal sacs were manually emptied during a DRE and an anal purse string suture was then placed. After standard aseptic preparation of the area, an inverted U-shape incision was performed between the base of the tail and the anus, starting from the ischiatic tuberosity up to the contralateral tuberosity.

The subcutaneous fat and perineal fascia were dissected to visualize the external anal sphincter, the rectococcygeal muscles and the dorsal rectum. The paired rectococcygeal and levator-ani muscles were transected, if necessary [9].

The mass was excised using a combination of sharp and blunt (even digital) dissection. Hemostasis was reached by a combination of vessel electrocautery and temporary gauze packing. If transected, the muscles were sutured with 2-0 or 3-0 absorbable monofilament. The subcutis and the skin were closed routinely [7].

Dorsal episiotomy was performed for vaginal masses. The dogs were placed in sternal recumbency [10] and prepared as previously described. The vaginal vestibule was lavaged with a diluted (1:10) saline 0.9%-povidone-iodine 1% solution. A urethral catheter was placed into the urethra, pre or intraoperatively in order to avoid iatrogenic damage. A skin incision was made starting from the dorsal commissure of the vulva toward the anus on the medial sagittal plane. A scalpel handle was inserted to assess the dorsal extent of the vestibule. The vestibular constrictor muscle and the mucosa were incised to visualize the mass. The tumor was excised with a combination of sharp, blunt and digital dissection. The surgical defect was closed in two layers (mucosa and submucosa and muscular vaginal tissue) using 2-0 or 3-0 absorbable monofilament; the subcutis and the skin were closed in a standard fashion [10].

For those tumors protruding in the abdomen or dropped cranial to the pelvis inlet during the perineal dissection, a standard midline caudal celiotomy was performed. Therefore, the dog was positioned in dorsal recumbency and the entire abdomen, previously clipped, was aseptically prepared. A median skin incision was performed from the umbilicus to the pubis [11,12]. The mass was excised using sharp and blunt dissection, if not previously carried out, i.e., in the case of sliding during the surgical perineal access (see before). Hemostasis was reached by electrocautery and temporary gauze packing.

After tumor removal, the descending colon, ureters and regional lymph nodes were inspected for abnormalities. The abdomen was closed routinely [11].

Intravenous fluid was usually continued for 24–48 h postoperatively according to the animal’s initial hydration status or until spontaneous eating and drinking. Analgesia consisted of opioids and non-steroidal anti-inflammatory drugs (NSAIDs).

Patients were generally discharged from the hospital 2–5 days after surgery with an Elizabethan collar and NSAID therapy. Antibiotic therapy was administrated 24 h postoperatively only if needed [13].

Postoperative complications and wound status were monitored until skin suture removal (usually 10 days after surgery). Long-term follow-up was obtained for all patients by telephone contact with the owners or referring veterinarians.

## 3. Results

### 3.1. Inclusion Criteria

Eleven dogs were eligible for inclusion in this study; each dog was identified by a number assigned at the beginning of the study.

### 3.2. Signalment and Clinical Findings

Mixed breeds were the most represented (64%). The breeds affected as well as the clinical signs at presentation are reported in Table 1.

Median age was 10 years (range 8–16.5; mean 11.1) and median weight was 20 kg (range 6–38; mean 2.5 kg). Eight dogs (72%) were females (6 intact and 2 spayed). There were 3 males (18%) (2 intact and 1 neutered). All the dogs suffered from fecal tenesmus. Four (36%) dogs were referred from the Critical Care Unit of the VTH for complete fecal obstruction (*n* 2, 9–11). Three (27%) dogs also had urinary problems (*n* 2, 6, 8), the urinary occlusion being complete in 2 cases (*n* 2, 6).

All the patients also had appetite loss, lethargy and abdominal pain. Other recorded clinical signs were weight loss and poor health conditions (*n* 2, 3, 6, 8, 9). At presentation, 3 patients (27%) had perineal swelling (*n* 3, 8, 9).

The median overall duration of clinical signs was 225 days (mean 217; range 90–365). Signalment, clinical signs and localization of the mass are reported in Table 1.

Digital rectal examination (plus vaginal examination in the case of a mass positioned ventrally to the rectum) was performed in all patients. A well-defined, compact, more or less mobile, mass was detected in all dogs. The mass was dorsal (or latero-dorsal) to the rectum in 8 dogs (72%) (*n* 1–6, 10, 11) and ventral to it in 3 dogs (18%) (*n* 7–9) (Table 1). In all the cases, a compact mass was palpated at DRE.

### 3.3. Diagnosis

A preoperative diagnosis was attempted in all cases. Ten patients (91%) underwent transcutaneous perineal (*n* 3, 9–11) or transrectal (*n* 1, 2, 4–7) fine needle aspiration (FNA) biopsy. An ultrasound-guided trans-abdominal tru-cut needle biopsy was performed in 1 case (9%) (*n* 8).

A preoperative cytological or histological diagnosis was obtained in 2 (18%) cases (*n* 1, 7) and 1 (9%) case (*n* 8), respectively. In the former case, the lesion was cytologically compatible with a soft tissue sarcoma (*n* 1) and a mesenchymal tumor (*n* 7). In the latter case, a sarcoma (*n* 8) was diagnosed by histology. Cytology was inconclusive in all the other cases (80%) (*n* 2–6, 9–11). (Table 1).

### 3.4. Pre-Surgical Imaging

Abdominal ultrasonography was performed in 8 dogs (72%) (*n* 1–8). In 3 cases (*n* 6–8) only the cranial part of the mass was visualized. In the remaining 5 dogs, the mass could not be seen.

Seven patients (63%) (*n* 3–5;8–11) underwent total body CT with intravenous (IV) contrast injection (Table 1).

Four patients (36%) (*n* 1, 2, 6, 7) had 3-view thoracic radiographs performed for staging purposes.

No dog had evidence of local (regional lymph nodes) or distant metastases.

Computed tomography revealed the presence of a single and 2 adjacent masses in 6 out of 7 (86%) (*n* 3–5, 9–11) and 1 out of 7 cases (14%) (*n* 8), respectively. The masses appeared well defined and homogeneous, with mild contrast enhancement in all cases. Colonic and rectal displacement, and urethral compression were always present (Figure 1).

High creatinine (1308 µmol/L [range 44–159]), blood urea nitrogen (214 mmol/L [range 6.1–21.1]) and phosphate (7.36 mmol/L [0.7–2.1]) serum levels due to a five-day urinary obstruction were present in one dog (*n* 6). The patient also had abdominal effusion (modified trasudate-exudate with 0.025 g/L protein concentration). Before proceeding with surgery, a Foley catheter was positioned and, after 2 days of IV hydration, the above values normalized, although they remained in the high range.

### 3.5. Surgery and Histological Evaluation

The surgical approach was chosen according to the localization of the mass.

A dorsal approach to the rectum was used in 6 cases (55%) (*n* 1–5, 10) (Figure 2).

A median caudal celiotomy was performed in 4 cases (36%) (*n* 6–8, 11). An episiotomy was performed in 1 dog (9%) (*n* 9). A double approach (perineal and abdominal) was used in 1 case (9%) (*n* 4) in which the mass, after being bluntly dissected using a dorsal approach to the rectum, slipped cranially into the abdominal cavity (Table 2).

The three patients (27%) having a vaginal mass underwent contextual ovariohysterectomy (*n* 7–9).

The tumor was easily removed principally using digital blunt dissection in all patients (91%) with the exception of one (*n* 9), in which the tumor was adherent to the urethra. The latter was damaged and subsequently repaired by means of an anastomosis at the level of the urethral papilla.

A firm, smooth, white/greyish and ovoid shaped or multilobulated mass was removed in all the dogs (Table 2). The largest median diameter was 6 cm (range 4–15; mean 9) (Table 2). After surgical removal, histopathological examination was carried out in all cases.

A well-differentiated leiomyosarcoma, a fibroleiomyoma, a fibroma and a leiomyoma were observed in 2 (18%) (*n* 3, 10), 1 (9%) (*n* 6), 3 (27%) (*n* 1, 8, 9) and 2 (18%) dogs (*n* 7, 11), respectively.

A benign mesenchymal neoplasm was histologically diagnosed in 3 (27%) cases (*n* 2, 4, 5); the histological findings were compatible with a fibroma or a leiomyoma; however, it was not possible to differentiate between them. Immunohistochemical investigations were proposed but were not accepted by the owners.

### 3.6. Surgical Complications and Outcome

Bleeding during the dissection of the mass occurred in all cases and was copious in some of them (*n* 3, 9), even though a blood transfusion was not needed in any case, and the bleeding was managed by combining both hemostasis with a electrocautery and gauze packing. During the surgical excision of the mass, the lumen of the rectum or the vagina was never penetrated.

No specific postoperative complications were observed, except for some swelling at the level of the perineum which did not require any treatment and spontaneously disappeared after 1 week. The surgical wounds healed uneventfully in all patients.

Bladder atony due to prolonged over-distension was observed in 3 (27%) patients (*n* 6, 8, 9). Two of these dogs (*n* 6, 8) developed urinary incontinence which spontaneously improved within two postoperative months, without, however, regaining complete functionality.

Immediately after surgery, functional urinary retention with anuria developed in 1 dog (*n* 9). A Foley catheter was placed and left in situ with a closed system collection for 5 days. Normal urination was restored within two postoperative weeks.

Fecal tenesmus resolved in all cases immediately after surgery.

The patients were discharged two days after surgery in all cases except for one (*n* 9) which was hospitalized until Foley catheter removal.

Follow-up was available for 7 patients (64%); four (36%) (*n* 1, 2, 7, 10) were lost to follow-up approximately 90 (*n* 1, 2, 10) and 1330 (*n* 7, at the last follow-up, the dog was in a healthy condition and tumor free) days after surgery. The latter dog was included in the ST evaluation because its follow-up time was longer than 1 year. Dogs (*n* 1, 2 and 10), lost in follow-up at approximately 90 days after surgery, were not included in the ST evaluation.

Owners, questioned about their dog’s quality of life after surgery stated that it was highly improved. No patient, at least for those which had a follow-up available, had a local recurrence, despite marginal excision of the tumor. Distant metastases were not observed in the 2 dogs (*n* 3, 10) with malignant disease.

Median follow-up time was 429 days (range 90–1825; mean 725). Median ST was 950 days (range 378–1825; mean 964) with overlapping of the DFI and the ST as no patient died from recurrence or metastasis.

At the time of writing, 2 out of the 7 (28%) dogs are still alive (*n* 6, 11), 3 out of the 7 (43%) dogs died from causes unrelated to the tumor (*n* 3 and n 4 from old age; n 9 from neurological problems the origin of which was not investigated), and 2 out of the 7 (28%) dogs developed other tumors (*n* 5-hepatic neoplasia 4 years after surgery; n 8-mammary carcinoma approximatively 1 year and a half after surgery).

## 4. Discussion

Large-sized, benign space-occupying pelvic masses are a rare finding in veterinary medicine, with only a few reports in the veterinary literature [6,7,14]. These slow-growing tumors are more often non-invasive, and arise from vaginal or rectal smooth muscle cells between the serosa or adventitia, and the submucosa [4,7]. Their behavior is usually benign, and marginal surgical excision can be an option, usually without post-surgical complications and with a fast recovery from the severe clinical signs at diagnosis. Prognosis is good after surgical removal [4,14,15]. These tumors may reach a large size before causing clinical signs due to their slow-growing and non-infiltrative behavior [7]. The extra-luminal growth within the non-distensible pelvic canal can be asymptomatic until a progressive compression of the rectum and/or the urethra occurs. Serious metabolic problems (uremia) and renal damage can be present, especially in the case of complete urethral occlusion. Finally, the compression can be so serious as to cause a complete urinary and/or fecal obstruction leading the animal to a life-threatening status.

Digital rectal examination, associated when needed with vaginal exploration, may provide important information regarding the non-invasive nature of the lesion, thus helping to differentiate between benign or malignant intra-pelvic tumors. At DRE, tumor length evaluation is not always possible, depending on the position of the mass in the pelvic canal and on its volume. Digital rectal examination also helps to define the lesion’s origin based on its position in relation to the rectum. If dorsal to it, the mass has to be differentiated from the sacral lymph nodes which are enlarged in the case of metastatic spread from perineal tumors. If ventral to the rectum, the mass has to be differentiated from a prostatic or vaginal mass, obviously depending on the sex. Urethral tumors do not usually have a mass-like appearance.

A firm, well defined mass, not completely (depending on tumor volume) attached to the surrounding structures, dorsal or ventral to the rectum, has an elevated possibility of being benign or behaving in a benign manner.

All the dogs in this retrospective study showed the findings described above at DRE. They were easily surgically removed, for the most part by digital blunt dissection, in all the patients, also including the two cases postoperatively histologically defined as well-differentiated malignant tumors. In the case of masses characterized by greater malignancy it probably would not have been possible to remove them so easily without leaving macroscopic portions of the tumor in place or without penetrating the lumen of the rectum or the vagina. Post-surgical histology confirmed the benign nature of the mass in 10 dogs and a low-grade leiomyosarcoma in 2 cases.

According to the literature, the dogs in this series were middle-aged [7,14]. Patients with a vaginal mass also underwent ovariohysterectomy to prevent local recurrence [16,17]. The clinical signs were mostly represented by fecal and urinary straining due to the colon-rectal and/or urethral compression.

In all cases, a preoperative diagnosis was attempted. Although a larger cellular specimen is expected from smooth muscle tumors as compared to other mesenchymal tumors, preoperative cytology was diagnostic in only 2 out of the 10 cases which underwent the procedure, likely due to the poor tendency of these tumors to exfoliate, [8]. However, given the low invasiveness of the procedure, FNA cytology should always also be performed to exclude the diagnosis of readily exfoliating malignant tumors, such as epithelial and round cell tumors [18].

A needle core biopsy via a tru-cut needle was attempted in 1 case and the histologic diagnosis of sarcoma was obtained. This procedure is more likely to yield a diagnosis than a fine needle biopsy; however, it has limitations due to its major invasiveness and the need for anesthesia. The waiting times for a histological report may also represent a limitation for those patients in which the urethra and the rectal mass compression could be so severe as to cause complete urinary and/or fecal obstruction, leading the animal to a life-threatening status.

Diagnostic imaging should always be carried out for staging purposes and for pre-surgical planning. Metastatic spread to the abdominal structures, dilation of the ureters and the renal pelvis, and abdominal effusion can be detected by abdominal ultrasound; the latter may not be very useful for visualizing intra-pelvic tumors. Osseous pelvic structures act as a barrier to a complete sonographic evaluation, and only the abdominal portion of the mass can be investigated. Nevertheless, in the case of a well-defined, firm, dorsally or ventrally located mass at DRE, abdominal ultrasound should be used as a first line diagnostic approach for investigating the urinary apparatus (and the prostate in male dogs) and the distal digestive tract, and for excluding ileo-sacral lymphadenopathy. 

Abdominal radiographs do not usually provide sufficient information regarding intrapelvic masses arising from soft tissues. Indirect signs due to the compression of the mass on the adjacent organs can be observed Abdominal radiographs can be useful for excluding pelvic bone involvement. Thoracic radiographs can be used to exclude lung metastases.

Computed tomography is currently the most indicated diagnostic tool in such cases since it is now easily accessible at relatively low cost. Compared with ultrasound and radiographic examination, CT provides information regarding the presence, location, size and extent of the disease, and its relation with the surrounding organs, thus providing information for surgical planning. Moreover, contrast enhancement better delineates the vessels into or in proximity of the mass. A previous study reported that benign masses, such as fibromas and leiomyomas, are more likely to have a homogeneous architecture after contrast administration due to less tissue necrosis and vascularization as compared with malignancies [1]. A CT scan is also useful for staging purposes.

Marginal surgery was performed on all the dogs in this series. Different approaches were chosen according to the localization of the mass. The perineal approach (dorsal to the rectum and episiotomy) is less invasive and has low morbidity; however, if the mass lies in the most cranial part of the pelvic canal or in the caudal abdomen, the perineal approach may not be sufficient for removal of the mass. In those cases, a median caudal celiotomy, or a combined perineal and abdominal approach may be required.

The dorsal approach to the rectum provides good surgical exposition of the caudal part of the rectum; since these tumors are often well encapsulated, a blunt dissection is generally enough to remove them entirely, preserving the structures involved in the fecal continence mechanism, i.e., the pelvic nerve plexus [8,19].

Intraoperative complications, including hemorrhage, are rare. In one dog (*n* 9) the urethra was iatrogenically damaged during the dissection; however, it was promptly replanted at the level of the vaginal urethral papilla. No other complications, such as infection and suture dehiscence, in the short-term follow-up, or fecal or urinary incontinence in the long-term follow-up were observed.

In addition, the dogs recovered early and the majority were discharged within 48 h with a rapid return to normal intestinal and urinary functionality.

One of the aims of this study was to point out how DRE represented a simple first clinical examination for differentiating between intra-pelvic malignant and benign masses. A well-defined firm mass, dorsal or ventral to the rectum originating from the outermost parts of the rectal or vaginal wall, can be clinically suspected at DRE. Its position in the cranial, central or caudal part of the pelvic canal can also be evaluated. A vaginal origin is suspected for a mass ventral to the rectum in a female dog. This was the case in the three intra-pelvic masses ventral to the rectum which were included in this retrospective study.

The clinical history of long lasting tenesmus together with the detection of a firm mass at DRE is surely not sufficient in itself for deciding to perform surgery without a preoperative diagnosis. An attempt to perform FNA of the mass should be carried out, despite the fact that it may not be diagnostic as previously described. Cytology negative for those tumors which exfolate more easily may also support the clinical hypothesis of a mesenchymal benign or well-differentiated intra-pelvic tumor.

A good option may be to perform a tru-cut needle biopsy when a CT scan is planned in order to reduce the number of anesthesia sessions in an already critical dog. As previously mentioned, the waiting times for a histological report may represent a limitation for those patients in serious clinical conditions due to the rectal and/or urethral compression of the mass.

In this study cytology was inconclusive in 80% of cases but a FNA was performed in all the dogs. It is the authors’ opinion that a clinical history of a long-lasting tenesmus, the detection of a large, firm mass at DRE, and a non-diagnostic cytology more likely in case of compact mesenchinal lesions, were sufficient to decide to proceed to surgery without a preoperative diagnosis. This decision was also supported by the fact that preoperative diagnostic imaging exams ruled out the involvement of other structures such as prostate and sacral lymph nodes and a metastatic tumor spread. Finally, to support this decision there were also the serious and life-threatening pre- surgical clinical conditions of some dogs.

Regarding the diagnostic imaging techniques most suitable in the case of clinically suspected benign or low-grade malignant intra-pelvic compressive masses, a total body CT scan actually represents the best choice. At CT, the evidence of a well-defined, homogenous mass, with mild contrast enhancement due to a limited vascular supply, in addition to no evidence of local and distant metastatic spread, may support the clinical hypothesis of a benign or of a low-grade malignant tumor [1].

In this study, CT was not performed in 4 cases due to the 18-year-long study period. At the beginning of the study period, this technique was not so commonly used in the field of veterinary medicine. For that reason, only abdominal ultrasound and chest radiographs were used in 4 cases. However, they were useful in excluding the possibility of tumor metastases and in highlighting both the mass, when positioned cranially in the pelvic cavity, and the indirect signs due to the compression of the mass on the adjacent organs.

This study had some limitations. The first was the low number of dogs included which could have weakened the conclusions to some extent; in addition, this study was retrospective, thus not all the data were available, such as the follow-up, since some dogs were lost to follow-up just a few weeks after surgery. No dog received imaging rechecks to exclude a local recurrence; however, clinical signs (tenesmus) suggesting a local recurrence were absent in all the dogs having an available follow-up. The 18-year-long study period also represented a limitation. The diagnostic procedures were different among the dogs included in the study.

## 5. Conclusions

Even if rare, intra-pelvic masses can be benign or behave in a benign manner. Clinical signs, referring to rectal and urethral compression and DRE can help the clinician in supporting this diagnostic hypothesis. Cytology is often inconclusive but helps to exclude epithelial or round cell malignant intra-pelvic masses. Total body CT represents the most useful diagnostic exam in such cases. All the dogs included in this study were referred to the VTH with the owners’ conviction of a poor prognosis since an unresectable malignancy was originally suspected by the referring veterinarians. In some cases, euthanasia had also been proposed. Despite this, after a thorough clinical examination, including a DRE combined with appropriate imaging exams, usually represented by a CT scan, a marginal excision was curative in all the dogs with an available follow-up, including those with a well-differentiated malignant tumor. Improvement in the quality of life was rapid in all cases.

## Figures and Tables

**Figure 1 animals-11-01361-f001:**
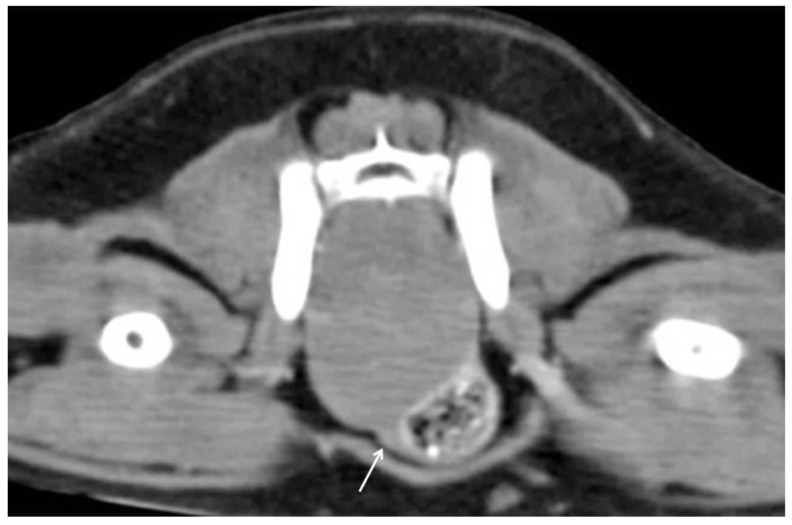
Tomographic appearance of a intra-pelvic mass. The colon is displaced ventrolaterally. The urethra is highlighted by the white arrow.

**Figure 2 animals-11-01361-f002:**
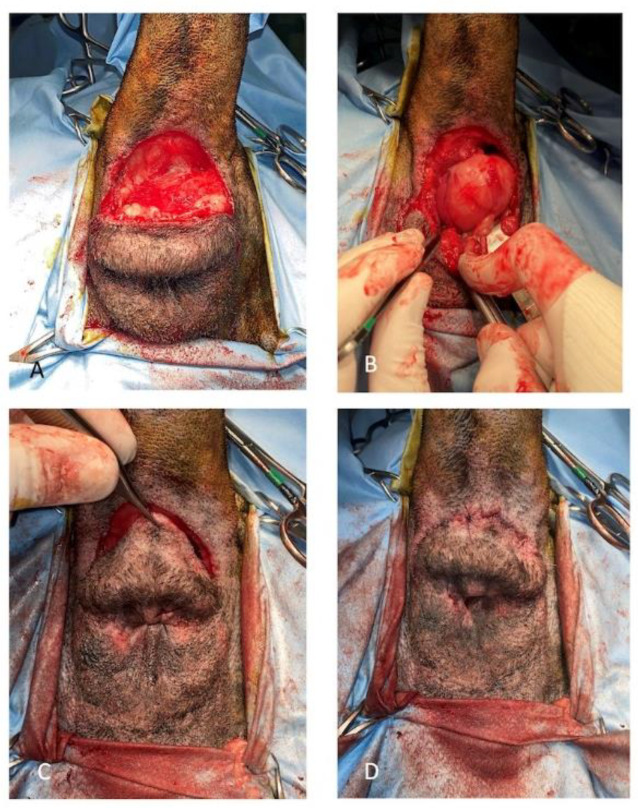
(**A**) Skin incision for the dorsal approach to the rectum. A purse string suture is positioned to occlude the anus. (**B**) Paired rectococcygeus and levator-ani muscles are transected. (**C**) The mass is gradually detached from the surrounding tissues using blunt and digital dissection. (**D**) Postoperative appearance of the perineum after the procedure (purse string removed).

**Table 1 animals-11-01361-t001:** Signalment, clinical signs, localization of the mass and diagnostic procedures.

Case Number	Breed	Weight (kg)	Age	Sex	Clinical Signs	Localization of the Mass	Imaging	Preoperative Diagnosis
1	Mixed	30	10	CM	Fecal tenesmus, loss of appetite, lethargy, abdominal pain	Dorsal to the rectum	Txr, AbUs	Y (cytology) STS
2	Mixed	13	9	F	Fecal and urinary tenesmus, complete fecal and urinary obstruction. loss of appetite, lethargy, abdominal pain	Lateral to the rectum	Txr, AbUs	N
3	Mixed	20	16,5	M	Fecal tenesmus, perineal swelling, loss of appetite, lethargy, abdominal pain	Dorsal to the rectum	AbUS, tbCT	N
4	Pomeranian Spitz	6	12	F	Fecal tenesmus, loss of appetite, lethargy, abdominal pain	Dorsal to the rectum	AbUS, tbCT	N
5	Mixed	20	13	SF	Fecal tenesmus, loss of appetite, lethargy, abdominal pain	Dorsal to the rectum	AbUS, tbCT	N
6	Rottweiler	38	9	SF	Fecal tenesmus, complete urinary obstruction, loss of appetite, lethargy, abdominal pain	Dorsal to the rectum	Txr, AbUs	N
7	Mixed	14	9	F	Fecal tenesmus, loss of appetite, lethargy, abdominal pain	Ventral to the rectum	Txr, AbUs	Y (cytology) mesenchymal tumor
8	English Setter	24	12	F	Fecal and urinary tenesmus, perineal swelling, loss of appetite, lethargy, abdominal pain	Ventral to the rectum	AbUS, tbCT	Y (histology) STS
9	Mixed	13	14	F	Fecal tenesmus, perineal swelling, complete fecal obstruction, loss of appetite, lethargy, abdominal pain	Ventral to the rectum	tbCT	N
10	Labrador Retriever	32	8	F	Fecal tenesmus. Complete fecal obstruc tion, loss of appetite, lethargy, abdominal pain	Dorsal to the rectum	tbCT	N
11	Mixed	27	10	CM	Fecal tenesmus. Complete fecal obstruction, loss of appetite, lethargy, abdominal pain	Dorsal to the rectum	tbCT	N

Txr: thoracic x-ray; AbUs: abdominal ultrasound; tbCT: total body computed tomography; STS: soft tissue sarcoma; SF: spayed female; CM: castrated male; Y: yes; N: no.

**Table 2 animals-11-01361-t002:** Aspect and histological features of the mass, the surgical procedures used, the surgical complications observed and outcome.

Case Number	Surgical Approach	Mass Aspect and Largest Diameter	Histology	Surgical Complications	Clinical Signs of Remission after Surgery	Recurrence/Metastasis	ST (Days)	Status
1	DtR	Ovoid shaped, 4,5 cm	Fibroma	N	Y	NA	90	Lost to FU
2	DtR	Ovoid shaped, 5 cm	Leiomyoma/Fibroma	N	Y	NA	90	Lost to FU
3	DtR	Multilobulated, 15 cm	Well differentiated leiomyosarcoma	N	Y	N	429	Died (not related to the tumor)
4	DtR + MeC	Ovoid shaped, 6 cm	Leiomyoma/Fibroma	N	Y	N	1825	Died (not related to the tumor)
5	DtR	Multilobulated, 4 cm	Leiomyoma/Fibroma	N	Y	N	1460	Died (hepatic neoplasia)
6	MeC	Ovoid shaped, 6 cm	Fibroleiomyoma	N	Y, but PM urinary incontinence	N	1335	Alive
7	MeC + OHE	Multilobulated, 10 cm	Leiomyoma	N	Y	N	1330	Lost to FU
8	MeC + OHE	2 ovoid shaped masses, 5 cm (total)	Multiple fibromas	N	Y, but PM urinary incontinence	N	570	Died (mammary carcinoma)
9	DE + OHE	Ovoid shaped, 12 cm	Fibroma	Y, urethral damage and reimplantation	Y, but transitory urinary incontinence	N	378	Died (neurological disease)
10	DtR	Ovoid shaped, 8 cm	Well-differentiated leiomyosarcoma	N	Y	NA	90	Lost to FU
11	MeC	Ovoid shaped, 15 cm	Leiomyoma	N	Y	N	382	Alive

DtR: dorsal to the rectum; LtR: lateral to the rectum; MeC: median celiotomy; DE: dorsal episiotomy; PM: permanent and mild; OHE: ovariohysterectomy; ST: survival time; NA: not available; FU: follow-up; Y: yes; N: no.

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
