# Peer review of "Benign or Low-Grade Malignant Masses Occupying the Pelvic Canal Space in 11 Dogs"

_animals, 2021, doi:10.3390/ani11051361_

Round 1

Reviewer 1 Report

After the changes, the work may be published. It is now clearer to the reader

Author Response

Thank you for accepting the manuscript for publication

Reviewer 2 Report

I would like to thank the authors for their efforts in addressing all the reviewer's comments.

Below I have a few comments for the revised manuscript. 

Line 98 I would replace the word "obtained" with "attempted"

Line 98-100 the sentence “The cytology in case of consistent mesenchymal lesions dorsal or ventral to the rectum is often not diagnostic but helps in differentiating these lesions from an epithelial or round cell lesion, more prone to exfoliate” lack reference

Line 100-101 "A biopsy for histological examination should be recommended in the case of non-diagnostic cytology" since it was not done in the present study, I would soften this sentence

Results section: Please be consistent when you refer to the number id of the patients i.e line 217 replace (n 7, 8, 9) with (n 7-9), line 233 replace (n 3;4;5;8-11) with (n 3-5, 8-11) etc...

Line 355-356 the total number of dogs is 12, probably only 9 had a benign tumor, please check your statement

Author Response

Rebuttal letter to Reviewer n° 2

As requested from the Reviewer the authors tried to improve the research design.

1) Reviewer comment line 98: I would replace the word "obtained" with "attempted"

 Authors answer: We thank the Reviewer for the comment. We replaced the word "obtained" with "attempted".

  2)      Reviewer comment line 98-100:   the sentence “The cytology in case of consistent mesenchymal lesions dorsal or ventral to the rectum is often not diagnostic but helps in differentiating these lesions from an epithelial or round cell lesion, more prone to exfoliate” lack reference

 Authors answer. We thank the Reviewer for the comment. We added the reference to the sentence (Friedrichs, K. R.; Young, K.M. Diagnostic Cytopathology in Clinical Oncology. In Withrow and MacEwen’s Small Animal Clinical Oncology, 6th ed.; Vail, D., Thamm, D., Liptak, J. Eds; Elsevier: Missouri, USA, 2020; pp 126-14).

3) Reviewer comment line 100-101: A biopsy for histological examination should be recommended in the case of non-diagnostic cytology" since it was not done in the present study, I would soften this sentence

Authors answer. We thank the Reviewer for the comment. We softened the sentence in the text, as suggested by the Reviewer.

4) Reviewer comment Results section. Please be consistent when you refer to the number id of the patients i.e line 217 replace (n 7, 8, 9) with (n 7-9), line 233 replace (n 3;4;5;8-11) with (n 3-5, 8-11) etc...

Authors answer. Thank you for the comment. We replaced commas with dashes.

5) Reviewer comment line 355-356 the total number of dogs is 12, probably only 9 had a benign tumor, please check your statement

Authors answer. We are sorry but we are not able to find in the text the statement reported by the reviewer.

The total number of dogs in the study is 11. Of those 11 dogs, 9 of them had a benign mass and only two had a low grade malignant mass (well differentiated leiomyosarcoma).

This manuscript is a resubmission of an earlier submission. The following is a list of the peer review reports and author responses from that submission.

Round 1

Reviewer 1 Report

Cancer is a serious problem in veterinary medicine. The work contains relevant content. The full course of modern medical procedure was presented. When reporting blood chemistry results, the authors should use SI units. In the materials and methods, it is worth adding what tomographic apparatus was used, as in the case of the ultarsonographic apparatus. My comments are included in the content of the publication. The treatment figures are appropriate.

Author Response

A point by point response to the reviewer's comment was made. The rebuttal letter to reviewer 1 is attached

Reviewer 2 Report

This 18-year long restrospective study evaluated clinical presentation, diagnostic findings and outcome of 11 dogs affected by intrapelvic masses treated with surgery alone. Even with the limitations of a long period retrospective study, the study findings could be very useful in clinical practice and could encourage other surgeons/vets to put efforts and positivity in the treatment of life-threating conditions as a rectal/urethral occlusive intrapelvic mass can be.

Line 32-34 The authors stated “…and diagnostic imaging are mandatory to assess…”, however as I can read, 4 of 11 dogs included in the study did not received CT or abd XR or other imaging to study the pelvic cavity. How can the authors conclude that diagnostic imaging is mandatory? I would soften conclusions. If the 3 dogs without pelvic imaging study are the oldest (no CT available), the authors could say “even if 3 out of 11 dogs had no diagnostic imaging for their pelvic masses, at today, in the authors’ opinion preoperative tridimensional diagnostic imaging is highly suggested/recommended” or similar

Line 103 I am not sure histology report can be associated with the word “behaviour”. Maybe the authors can write “histology report confirming benign or low grade/well-differentiated malignant tumour”.

Line 120 Among ST, did you include death for any reason or only tumour-related death? Please define

Line 133 Please replace all the comma with dots through the manuscript and tables i.e 0,2 mg/kg with 0.2 mg/kg

Line 124-184 These parts sound like the study was prospective. For an 18-year long study period, how can the authors assure all the procedures were performed in the same way? I am sorry, but it doesn’t sound realistic. Moreover, when I have read the result section and tables, I found the authors described very well their findings, with some missed information as usually happened in retrospective studies. I would suggest the authors to use words like “generally, frequently, routinely, usually” to tell the reader they usually (not always) do like this. The authors can also add to table 1or 2 a retrospective evaluation of the vary anaesthesiologic/analgesic protocols if considered important/substantial for this manuscript.

Line 145 The word “the” is repeated

Line 189 see the comment above regarding the word “behavior”

Line 191 Please remove spaces between 6 and 38

Line 217-220 I would move digital and vaginal examination procedures in the M&M section

Result section sounds in line with a retrospective study

Figure 1 Please if possible, tell the reader where the urethra is in this picture.

Table 2 Column status, dog 5: hepatic neoplasia lack a bracket; dog 7 was lost in FU, was recurrence status N corrected? Legend: FU: follow up. I am not a pathologist but I guess grading of STS is referring to cutaneous and subcutaneous STS, probably well-differentiated leiomyosarcoma is more appropriate than leiomyosarcoma 1st grade

Line 283 Please state what the authors mean with copious. Did it need blood transfusion? Did it change postHct? If yes, how much did hct decrease? Maybe authors can better define in the M&M section how they assessed/scored intra- (i.e hemorrage entity, rectal/urethral damage, ecc…) and post- operative complications (perineal hernia, urinary/fecal incontinence, …).

Discussion section

I would like to read discussions regarding the authors choice to perform surgery even if they lack of preoperative diagnosis in almost all cases and CT in 4 of 11 cases. Moreover I would like to know if their choice to marginally excised well-differentiated leiomyosarcoma is advisable or not. When I have finished to read the discussion section I understand that cytology is always recommended (but in the present study only 20% of samples were adequate), advanced diagnostic imaging should be always performed (but 4/11 cases did not have a CT), preoperative differentiation between malignant and benign tumors is very important to address surgery (but it was not available in many of these patients). I would suggest the authors to discuss more about what they did instead of what they could do. I think this paper has some important findings and can push other surgeons to put efforts and positivity in the treatment of this life-threating condition even if an underneath malignant tumor has chance to be.

Line 314 Please replace “lare size” with “large-sized”?

Line 315 Please replace “this” with “these”

Line 321-323 I apologize but I do not understand how the authors could say tumors arose from the rectal wall. Did rectal leiomyoma/sarcomas presented as sessile or peduncolated masses? As I can read the authors have never needed to perform a colotomy/colectomy. Have the authors never needed to cut the rectal mucosa? How can the authors assure these tumors arose from the rectal wall and not from the intrapelvic perirectal soft tissues? Did the authors use adjuvant therapy for marginal resected leiomyosarc?

Line 336 life-threating

Line 340 “(n=3)” is not necessary.

Line 341 In result section you said cytology was performed in 10 patients (line 240-241)

Line 340-356 In the end the authors went in surgery without a diagnosis in the majority of cases, confirming the likely uselessness of cytology. Moreover, cytology failed to diagnose both of the leiomyosarcomas.

Line 358-360 In my opinion, the statement “Abdominal XR do not provide …” it is questionable because pelvic masses originating from the bone i.e MLO can also see with abdXR changing surgical procedure

Line 374-377 “Distinction between a malignant and a benign pelvic tumor is very important”, however the authors did not have a preoperative diagnosis in almost of the cases and decided to use marginal surgery. Please discuss this incongruent thought

Line 341 please replace “in only in two dogs” with “in only two dogs”

Line 355 Please change “hoever”

Line 395-397 Please add in the limitations that no dogs received imaging rechecks to exclude local recurrence.

Conclusions

Based on their findings, the authors can give the reader advices/suggestions regarding essential preoperative diagnostic procedures and surgical approaches useful to obtain similar good outcomes.

Author Response

A point by point response to the reviewer's comments was made. The rebuttal letter to reviewer 2 is attached

Thank you

Reviewer 3 Report

Although this manuscript is of general interest for the readers since it reinforces the knowledge regarding the surgical possibilities for dogs with neoplastic masses in pelvic canal, which are often considered poor surgical candidates by many clinicians, it needs to be re-write before considered for publication.

It is poorly written, with some loose and meaningless sentences. It contains several grammatical and misspelling errors (ex. defectation, istology, lesion’s, hoever) and needs an extensive English revision.

The title does not fully translate the work. The introduction is poorly organised and does not significantly contextualise the interest of this study. The study methodology is not robust enough: the inclusion criteria mention only benign or low-grade malignant tumours, without explaining why the other tumours were rejected (and there are no exclusion criteria). In addition, the authors mention that these dogs were followed to determine also metastatic disease, but they are selecting mostly benign tumors… which does not metastasize by definition. The results section is poorly organised; the authors need to make an effort to describe cases grouping the findings in a meaningful way (and also presenting percentages in addition to the total numbers) allowing the reader to understand their purpose and to draw conclusions. In the discussion, please avoid loose and meaningless sentences such as line 357. This section should interpret and describe the significance of the findings in light of what is already known about the research problem and should explain any new understanding or insights that emerged as a result of the study problem. The discussion should be connected to the introduction by the research questions or hypotheses posed and may clearly explain how the study advanced the reader's understanding of the research problem that motivated this manuscript.

Author Response

A point by point response to the reviewer's comments was made. The rebuttal letter to reviewer3 is attached

Thank you
